# Dynamic Graph Unlearning: A General and Efficient Post-Processing Method via Gradient Transformation

## ABSTRACT

Dynamic graph neural networks (DGNNs) have emerged and been widely deployed in various web applications (e.g., Reddit) to serve users (e.g., personalized content delivery) due to their remarkable ability to learn from complex and dynamic user interaction data. Despite benefiting from high-quality services, users have raised privacy concerns, such as misuse of personal data (e.g., dynamic user-user/item interaction) for model training, requiring DGNNs to "forget" their data to meet AI governance laws (e.g., the "right to be forgotten" in GDPR). However, current static graph unlearning studies cannot *unlearn dynamic graph elements* and exhibit limitations such as the model-specific design or reliance on pre-processing, which disenable their practicability in dynamic graph unlearning. To this end, we study the dynamic graph unlearning for the first time and propose an *effective*, *efficient*, *general*, and *post-processing* method to implement DGNN unlearning. Specifically, we first formulate dynamic graph unlearning in the context of continuous-time dynamic graphs, and then propose a method called Gradient Transformation that directly maps the unlearning request to the desired parameter update. Comprehensive evaluations on six real-world datasets and state-of-the-art DGNN backbones demonstrate its effectiveness (e.g., limited drop or obvious improvement in utility) and efficiency (e.g., 7.23× speed-up) advantages. Additionally, our method has the potential to handle future unlearning requests with significant performance gains (e.g., 32.59× speed-up).

## CCS CONCEPTS

• **Security and privacy** → **Web application security**; • **Computing methodologies** → **Neural networks**.

## KEYWORDS

Dynamic Graphs, Unlearning, Privacy, GNN, Trustworthiness.

**ACM Reference Format:**
Anonymous Author(s). 2018. Dynamic Graph Unlearning: A General and Efficient Post-Processing Method via Gradient Transformation. In *Proceedings of Make sure to enter the correct conference title from your rights confirmation emai (Conference acronym 'XX).* ACM, New York, NY, USA, 14 pages. https://doi.org/XXXXXXX.XXXXXXX

## 1 INTRODUCTION

Dynamic graphs represent entities as nodes and their temporal interactions as dynamic links, making them suitable for modeling data in various web applications such as social networks [1] and trade networks [2]. Excelling in capturing temporal information in dynamic graphs [3], dynamic graph neural networks (DGNNs) [4] have recently emerged and proven to be effective in serving users in real-world applications, which have substantially improved user experiences in daily life. For example, in Reddit (i.e., a social news platform) [5], considering users and items (e.g., Reddit posts) as nodes and their temporal interactions as dynamic edges [6], the well-trained models [7] can be used as recommender systems to offer personalized content delivery services to users [8, 9].

Despite service benefits, users have raised concerns about the potential privacy issues associated with their sensitive data in web applications (e.g., interaction history with other users in Reddit [10]). For example, web applications might collect user data—either intentionally or occasionally—without user authorization to train service models [11], as these models require vast and comprehensive data for optimal functionality [12]. Therefore, when users issue requests for deleting data from web applications, they expect that corresponding DGNNs forget the knowledge acquired from their data, which is also in compliance with laws for AI governance (e.g., the "right to be forgotten" in GDPR [13]).

To address the privacy concerns of users above, graph unlearning [14, 15] is steadily attracting the attention of academic researchers and industry professionals. However, existing approaches designed for static graph unlearning and are less suitable for dynamic graph unlearning. For example, these methods (e.g., SISA [16]) can only unlearn static graph elements (e.g., static edges) and become ineffective when **unlearning dynamic graph elements** (e.g., dynamic edge events), which are vital components in dynamic graphs. Moreover, these static graph unlearning methods have limitations such as (1) *reliance on pre-processing*, (2) *model-specific design*, (3) *impractical resource requirements*, (4) *changing model architecture*, or (5) *overfitting unlearning samples*. (Refer to Section 2 for more details).

To this end, we propose the design of an *effective*, *efficient*, *general*, and *post-processing* dynamic graph unlearning method, called *Gradient Transformation*, which is suitable for DGNN unlearning. Specifically, we first define the dynamic graph unlearning request as the unlearning of a set of unordered dynamic events, followed by formulating the design goal of dynamic graph unlearning. In our approach, we consider the target DGNN model as a tool at hand that natively handles the dynamic graph and target model architecture during the unlearning process. To overcome limitations (1) (2) (4) of existing graph unlearning methods, we propose a gradient-based post-processing model to obtain the desired parameter updates w.r.t. unlearning of DGNNs, without changing the architecture of target DGNN models. With specially designed architecture and loss

functions, our unlearning paradigm avoids the requiring impractical resource issue (i.e., limitation (3)) and obviously alleviates the overfitting of fine-tuning methods (i.e., limitation (5)), respectively. Table 1 shows the differences between our method and typical graph unlearning methods, and our contributions in this paper are summarized as follows:

- For the first time, we study the unlearning problem in the context of dynamic graphs, and formally define the unlearning requests and design goal of dynamic graph neural network unlearning.
- We propose a novel method "Gradient Transformation" with a specialized loss function, which duly handles the intricacies of unlearning requests, remaining data, and DGNNs in the dynamic graph unlearning process.
- We empirically compare our approach with baseline methods on six real-world datasets, evaluations demonstrate the outperformance of our method in terms of *effectiveness* and *efficiency*.
- In addition to the *post-processing*, *architecture-invariant*, and *general* characteristics, our learning-based unlearning paradigm potentially highlights a new avenue of graph unlearning study.

## 2 RELATED WORK

For machine unlearning, a developer could ideally retrain a new model on updated data excluding the unlearning data [15]. However, retraining incurs significant resource costs, particularly with complex architectures and large datasets. Therefore, several methods have been proposed to efficiently unlearn graph data [17].

Current unlearning methods for graph data include the SISA method, influence function-based methods, the fine-tuning method, and other methods [17]. **(1)** The SISA method [16, 18] splits the training data into subsets, training a submodel on each. These submodels are combined to serve users. In response to an unlearning request, the developer identifies the subset with the data in the request, removes it, and retrains the corresponding submodel. **(2)** Influence function-based methods examine the impact of a training sample on the model parameters [19, 20]. When there are unlearning requests, they compute the gradients of these samples and the Hessian matrix of the target model to estimate the parameter update needed for unlearning. **(3)** The fine-tuning method, like GraphGuard [12], adjusts the model parameters by increasing the loss on unlearning samples and decreasing it on others, which facilitates model unlearning by lowering the accuracy on unlearning samples. **(4)** In addition to the methods mentioned above, alternative techniques (e.g., the Projector [21] for linear GNNs) are also applicable for graph unlearning. Refer to Appx. B.2 for more details.

As shown in Table 1, due to the following limitations of the static graph unlearning methods above, they are not suitable for the unlearning of dynamic graph neural networks. **(1)** *Reliance on pre-processing*. The SISA method can only be used in the initial development stage. Considering that many DGNNs have been deployed to serve users [27], SISA methods cannot be adapted for post-processing unlearning of these DGNNs, limiting their practicability and applications. **(2)** *Model-specific design*. For influence function-based methods, their parameter update estimations are generally established for simple GNN models (e.g., linear GCN model in CEU [25]), whose estimation accuracy is potentially limited due to the complexity and diversity of current state-of-the-art

**Table 1: Comparison between our method and typical studies.**

| Methods | Graph Type | Attributes | | |
|---|---|---|---|---|
| | | General | Post-Processing | Architecture Invariant |
| GraphEraser [16] | S | ● | ○ | ● |
| RecEraser [18] | S | ● | ○ | ● |
| GUIDE [22] | S | ● | ○ | ● |
| CGU [23] | S | ○ | ● | ● |
| GIF [24] | S | ◑ | ● | ● |
| CEU [25] | S | ◑ | ● | ● |
| Projector [21] | S | ○ | ● | ● |
| GNNDelete [26] | S | ● | ● | ○ |
| GraphGuard [12] | S | ● | ● | ● |
| **Ours** | D | ● | ● | ● |

* In this table, "S" and "D" denotes the static and dynamic graph, respectively. ○ indicates "Not covered", ◑ indicates "Partially covered", ● indicates "Fully covered". "General" indicates if one method can be used in the unlearning of a wide range of different model architectures. Refer to Appx. B.2 for more details on the principle and limitations of existing graph unlearning methods.

DGNN architectures [28]. Furthermore, these methods are potentially sensitive to the number of GNN layers (e.g., GIF [24]), which further makes these methods dependent on the target model architecture. **(3)** *Impractical resource requirements*. Although influence function-based methods can be used in a post-processing manner, calculations about the Hessian matrix of parameters typically incur high memory costs, particularly when the size of the model parameters is large. For instance, unlearning a state-of-the-art DGNN model named DyGFomer (comprising 4.146 MB floating point parameters) demands a substantial 4.298 TB of storage space, far surpassing the capabilities of common computational (GPU) resources. **(4)** *Changing model architecture*. Some methods (e.g., GNNDelete [26]) introduce additional neural layers to target models, which makes them impractical in scenarios where the device space for model deployment is limited (e.g., edge computing devices [29]). **(5)** *Overfitting unlearning samples*. The fine-tuning method (e.g., GraphGuard [12]) optimizes the parameter of the target model in a post-processing manner. However, the fine-tuning method is prone to overfitting unlearning samples, which potentially harms the model performance on remaining data [30].

## 3 PRELIMINARIES

Given a time point $t$, a static graph $G$ at time $t$ comprises a node set $\mathcal{V} = \{v_1, \ldots, v_{|\mathcal{V}|}\}$ and an edge set $\mathcal{E} = \{\ldots, (v_i, v_j), \ldots\}$, which delineates the relational structure among nodes. Generally, this static graph can be expressed as $G = (\mathbf{A}, \mathbf{X})$, where $\mathbf{X} \in \mathbb{R}^{|\mathcal{V}| \times d}$ ($d$ indicates the dimensionality of node features) and $\mathbf{A}_{i,j} = 1$ if $e_{ij} = (v_i, v_j) \in \mathcal{E}$, otherwise $\mathbf{A}_{i,j} = 0$. To investigate the change of graph data over time in some practical scenarios, numerous studies have emerged to explore dynamic graphs and capture their evolving patterns during a time period [31]. Depending on the characterization manner, dynamic graphs can be categorized into *discrete-time dynamic graphs* and *continuous-time dynamic graphs* [27]. In this paper, we focus on continuous-time dynamic graphs. The definition of discrete-time dynamic graphs can be found in the Appx. A.

**Continuous-time Dynamic Graphs (CTDG).** Given an initial static graph $G_0$ and a series of events $O$, a continuous-time dynamic graph is defined as $S = \{G_0, O\}$, where $O = [o_1, \ldots, o_N]$ represents a sequence of observations on graph update events. For example,

an event observation sequence with size 3 (i.e., $N = 3$) could be $O = [(add\ edge, (v_1, v_3),\ 24\text{-}Dec\text{-}2023), (add\ node, v_6, 25\text{-}Dec\text{-}2023), (Feature\ update, (v_2, [1, 0, 1, 0]),\ 25\text{-}Dec\text{-}2023)]$, where each event $o$ indicates an observation of graph updates. For example, $o_1$ indicates that an edge between $v_1$ and $v_3$ is added to $G_0$ on $24\text{-}Dec\text{-}2023$. Note that the symbol $[\cdot]$ in $O$ indicates that the *event order* is vital for $S$, as the following dynamic graph neural networks are designed to learn the temporal evolution patterns included in $S$.

**Dynamic Graph Neural Networks (DGNNs).** For static graphs, various methods (e.g., GCN [32]) have been proposed to capture both node features and graph structures. In addition to these two aspects of information, DGNNs are devised to learn the additional and complex temporal changes in dynamic graphs. For example, in DGNNs [27], the message function designed for an edge event (e.g., $(add\ edge, (v_i, v_j), t)$) could be

$$\begin{aligned}\mathbf{m}_i &= msg(\mathbf{s}_i(t^-), \mathbf{s}_j(t^-), \Delta t, \mathbf{e}_{i,j}(t)), \\ \mathbf{m}_j &= msg(\mathbf{s}_j(t^-), \mathbf{s}_i(t^-), \Delta t, \mathbf{e}_{i,j}(t)),\end{aligned} \quad (1)$$

where $\mathbf{s}_i(t^-)$ represents the latest embedding of $v_i$ before time $t$, $\Delta t$ indicates the time lag between current and last events on $(v_i, v_j)$, and $\mathbf{e}_{i,j}(t)$ denotes possible edge features in this event. Following this way, various dynamic graph neural networks have been proposed to obtain node representations that capture spatial and temporal information in dynamic graphs [33].

**Tasks on Dynamic Graphs.** Two common tasks on dynamic graphs are node classification and link prediction. Generally, given a dynamic graph $S$, a DGNN serves as the encoder to obtain the embedding of nodes, followed by a decoder (e.g., MLP) to complete downstream tasks. Here, we use $f$ to denote the entire neural network that maps a dynamic graph $S$ to the output space desired by the task. In *node classification* tasks, the parameter $\theta$ of $f$ is trained to predict the label of the nodes at time $t$. In *link prediction* tasks, $f$ is trained to infer if there is an edge between any two nodes $v_i$ and $v_j$ at time $t$. Taking the link predictions as an example, the optimal parameter $\theta^*$ is obtained by

$$\theta^* = \mathcal{A}_f(S) = \text{argmin}_\theta\ \ell\left(\mathbf{A}_t \mid f_\theta(S)\right), \quad (2)$$

where $\mathcal{A}_f$ is the algorithm (e.g., Gradient Descent Method) used to obtain the optimal parameter $\theta^*$ of $f$, $\mathbf{A}_t$ denotes the true adjacency matrix at time $t$, and $\ell$ represents a loss function (e.g., cross entropy loss). After *training from scratch* with $\mathcal{A}_f$ and $S$ ($t \leq T_S$), the $f_{\theta^*}$ could be used to make predictions in the future (i.e., $t > T_S$), where $T_S$ indicates the maximum event time in $S$.

## 4  PROBLEM FORMULATION

As shown in Figure 1, a DGNN $f_{\theta^*}$ is optimized to learn from its training dataset $S$ ($T_S = 8$) and then serve users. When a small amount of data $S_{ul}$ ($S_{ul} \subset S$) needs to be removed from the original training dataset (e.g., privacy concerns [14]), unlearning requires that the model $f_{\theta^*}$ optimized on $S$ must be updated to forget the knowledge learned from $S_{ul}$. Specifically, an unlearning method $\mathcal{U}$ aims to update the parameter of $f$ to satisfy

$$\text{dis}(P(\mathcal{U}(\mathcal{A}_f(S), S, S_{ul})), P(\mathcal{A}_f(S_{re}))) = 0, \quad (3)$$

where $\text{dis}(\cdot, \cdot)$ denotes a distance measurement, $P(\cdot)$ indicates the parameter distribution. $\mathcal{A}_f(S_{re})$ represents retraining from scratch

with the remaining dataset $S_{re}$, where $S_{re} = S \setminus S_{ul}$, $S = S_{re} \cup S_{ul}$, and $S_{re} \cap S_{ul} = \emptyset$.

**Unlearning Requests.** To clarify the unlearning of DGNNs, here we define the common unlearning requests. Note that the CTDG definition in Section 3 demonstrates the conceptual capability to capture spatial-temporal changes in dynamic graphs [27], while real-world datasets for dynamic link prediction are commonly only composed by adding edge events such as $(add\ edge, (v_1, v_5),\ 26\text{-}Dec\text{-}2023)$ [2]. Taking into account this fact, we propose the following definition of **edge unlearning requests**. An edge unlearning request $S_{ul}$ is made up of edge-related events that come from the original training data $S$. For example, $S_{ul} = \{(add\ edge, (v_1, v_3),\ 24\text{-}Dec\text{-}2023), (add\ edge, (v_1, v_5),\ 26\text{-}Dec\text{-}2023), (add\ edge, (v_2, v_5),\ 25\text{-}Dec\text{-}2023)\}$ requests to forget 3 different "$add\ edge$" events. $\{\cdot\}$ suggests that $S_{ul}$ is not sensitive to the event order, as $S_{ul}$ is only used to indicate which events should be forgotten. Edge unlearning requests frequently arise in practical scenarios. For instance, once their issues are resolved, users may wish to retract negative product reviews [3] that have influenced personalized recommendations.

*Remark.* Additional types of unlearning request may also occur in DGNNs. For example, $S_{ul}$ is called a *node unlearning request* if all events in it are related to all activities of specific nodes in the original training dataset $S$. For example, node unlearning requests potentially occur in cases where users arise de-registration or migration requests in a platform (e.g., Reddit [34]), where their historical interactions (i.e., edge events) are expected to be forgotten. Note that, considering that current real-world datasets for dynamic link prediction are commonly only composed of edge events, the above node unlearning request can be transferred into the corresponding unlearning edge events. Moreover, current practical datasets typically only include adding edge events [35]. Therefore, this study addresses typical unlearning requests related to adding edge events.

**Design Goal.** Our objective is to design an unlearning method $\mathcal{U}$ as follows. Given a dynamic graph $S$, a DGNN $f_{\theta^*}$ trained on it, and an unlearning request $S_{ul}$, the unlearning method $\mathcal{U}$ takes them as input and outputs the desired parameter update of $f$, i.e., $\Delta\theta = \mathcal{U}_\varphi\left(\mathcal{A}_f(S), S, S_{ul}\right)$. The $\varphi$ indicates the parameter of $\mathcal{U}$, and it is expected to satisfy

$$\min_\varphi \text{dis}(f_{(\theta^* + \Delta\theta)}, f_{\theta_{ul}^*}), \quad (4)$$

where $\theta_{ul}^* = \mathcal{A}_f(S_{re})$ indicates the ideal parameter and $\theta^* + \Delta\theta$ represents the estimated parameter.

## 5  GRADIENT TRANSFORMATION FOR UNLEARNING

By analyzing $\Delta\theta = \mathcal{U}\left(\mathcal{A}_f(S), S, S_{ul}\right)$, we find that the inputs of $\mathcal{U}$ play different roles in the process of unlearning. Generally, $S_{ul}$ issues unlearning requests, $S_{ul}$ and $\mathcal{A}_f$ determine the direction of parameter update w.r.t. unlearning. However, directly applying the parameter update derived only from $S_{ul}$ to $f_{\theta^*}$ will generally lead to overfitting on $S_{ul}$ and huge performance drops on $S_{re}$. Therefore, $S_{re}$ and $\mathcal{A}_f$ work together to improve the direction of parameter update, avoiding the adverse effects of using only $S_{ul}$. This role analysis suggests that there is a transformation which maps the initial gradient to the ultimate parameter updates during the unlearning.

                                                            

**Figure 1: An overview of the unlearning of DGNNs. (1) In the upper half, given a dynamic graph $S$ and a DGNN $f$, the model developer obtains the optimal $f_{\theta^*}$, which makes accurate predictions on both training ($t \leq 8$) and test ($t > 8$) data. (2) The lower half illustrates the ideal unlearning process. Upon receiving the unlearning request $S_{ul}$, the model developer removes $S_{ul}$ from $S$ and retrains $f$ from scratch to obtain the ideal DGNN $f_{\theta^*_{ul}}$. However, retraining from scratch requires huge source costs (e.g., time and computational source). (3) To this end, this paper aims to devise an *effective* and *efficient* unlearning method $\mathcal{U}$ to approximate the parameter obtained from retraining, as indicated by the green arrows.**

**Overview**. Given the above intuition, we propose a method called *Gradient Transformation* to implement dynamic graph unlearning. (1) For the mapping process, we take the initial gradient $\nabla \theta$ w.r.t. $S_{ul}$ as input and transform it with a two-layer MLP-Mixer model to obtain the desired parameter update $\Delta\theta$, i.e., $\mathcal{U}_\varphi : \nabla\theta \to \Delta\theta$. (2) Note that the ideal parameter $\theta^*_{ul}$ is not known by $\mathcal{U}_\varphi$ in Eq. 4. Thus, we propose a loss function to simulate the desired prediction behaviors of an ideal model $f_{\theta^*_{ul}}$.

**Gradient Transformation model**. Given a dynamic graph $S$, the optimal parameter $\theta^*$ of a DGNN $f$ is obtained by solving $\theta^* = \arg\min_\theta \ell(Y \mid f_\theta(S))$. As shown in Figure 2, after receiving unlearning requests $S_{ul}$, we first calculate the corresponding gradient $\nabla \theta$ w.r.t. the desired unlearning goal. Specifically,

$$\nabla\theta = \frac{d}{d\theta}\ell(\widehat{Y_{ul}} \mid f_{\theta^*}, S_{ul}), \tag{5}$$

where $\widehat{Y_{ul}}$ indicates the desired prediction results on the unlearning samples $S_{ul}$. For example, in the context of link prediction tasks and given $S_{ul} = \{o_1, ..., o_n\}$ consists of "add edge" events, the initial gradient information is obtained by $\nabla\theta = \frac{d}{d\theta}\sum_{o \in S_{ul}}\ell(\mathbf{A}_{t;i,j} = 0 \mid f_{\theta^*})$, where $i$, $j$ and $t$ represent the nodes and time involved in the event $o = (add\ edge, (v_i, v_j), t)$. $\mathbf{A}_{t;i,j} = 0$ indicates that $f$, which predicts $\mathbf{A}_{t;i,j} = 1$, is expected to forget the existence of $S_{ul}$ in its original training data $S$.

Given the initial gradient $\nabla\theta$, our gradient transformation model $\mathcal{U}$ takes it as input and outputs the desired parameter update $\Delta\theta$ of the target model $f$. Combined with the original parameter $\theta^*$, the updated parameter is obtained by applying $\theta^* + \Delta\theta = \theta^* + \mathcal{U}(\Delta\theta)$,

where $f_{\theta^* + \Delta\theta}$ is expected to behave the same as $f_{\theta^*_{ul}}$. In this paper, we use a two-layer MLP-Mixer to serve as the unlearning model $\mathcal{U}$. As shown in Figure 2, given the initial gradient as (i.e., $\nabla\theta \to \mathbf{H}_{in}$), the operation of $\mathcal{U}$ is presented as follows

$$\mathbf{H}^{(1)}_{tok} = \mathbf{H}_{in} + \mathbf{W}^{(2)}_{tok}\mathsf{GeLU}\left(\mathbf{W}^{(1)}_{tok}\mathsf{LN}(\mathbf{H}_{in})\right),$$

$$\mathbf{H}^{(1)}_{cha} = \mathbf{H}^{(1)}_{tok} + \mathsf{GeLU}\left(\mathsf{LN}(\mathbf{H}^{(1)}_{tok})\mathbf{W}^{(1)}_{cha}\right)\mathbf{W}^{(2)}_{cha},$$

$$\mathbf{H}^{(2)}_{tok} = \mathbf{H}^{(1)}_{cha} + \mathbf{W}^{(4)}_{tok}\mathsf{GeLU}\left(\mathbf{W}^{(3)}_{tok}\mathsf{LN}(\mathbf{H}^{(1)}_{cha})\right), \tag{6}$$

$$\mathbf{H}^{(2)}_{cha} = \mathbf{H}^{(2)}_{tok} + \mathsf{GeLU}\left(\mathsf{LN}(\mathbf{H}^{(2)}_{tok})\mathbf{W}^{(3)}_{cha}\right)\mathbf{W}^{(4)}_{cha},$$

where GeLU indicates the active function and LN denotes the layer normalization. With $\Delta\theta = \mathcal{U}(\Delta\theta)$ (i.e., $\Delta\theta = \mathbf{H}^{(2)}_{cha}$), we obtain the unlearned model $f_{\theta^* + \Delta\theta}$. In this paper, all parameters of Eq. (6) are denoted as $\varphi$, and $\mathcal{U}_\varphi$ is optimized by the following loss function.

**Unlearning Loss Function**. Unlearning $S_{ul}$ has the following potential impacts on DGNN $f_{\theta^*}$, and the corresponding loss functions describe the desired behaviors of $f_{\theta^*_{ul}}$.

**(1)** *Changed predictions on invariant representations.* An event $o_i$ in $S_{ul}$ could have observed a series of events that remained in $S_{re}$ before its occurrence time, while the unlearning expects $f$ to predict differently on the same representations. Therefore, the unlearning loss is defined as

$$\ell_{ul} = \ell(\widehat{Y_{ul}} \mid f_{\theta^* + \Delta\theta}, S_{ul}), \tag{7}$$

**(2)** *Invariant predictions on changed representations.* For an even $o \in S_{re}$, its involved nodes may have different representations

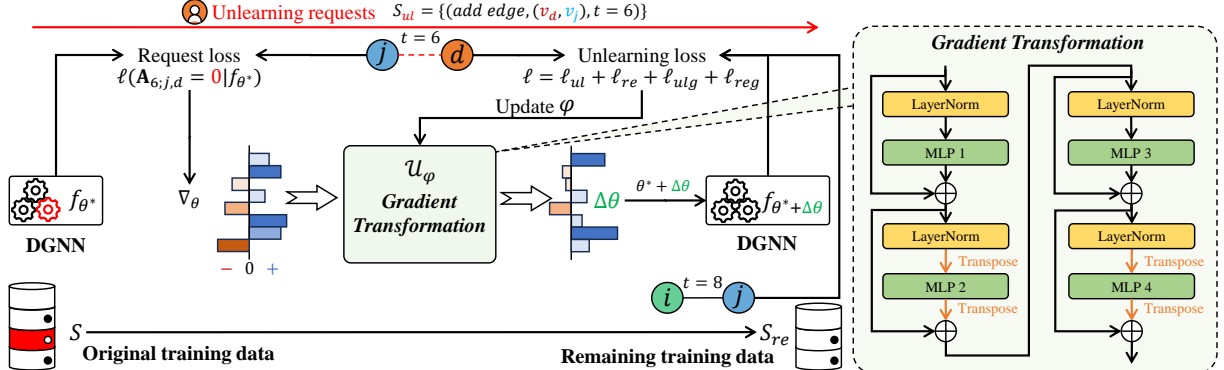

**Figure 2: The overview of our *Gradient Transformation* method.**

because their neighbors are potentially different before and after removing $S_{ul}$, while it requires $f$ to make unchanged predictions (as shown by $(v_i, v_j)$ at $t = 8$ in Figure 4). Thus, the performance loss for $S_{re}$ is defined as

$$\ell_{re} = \ell(Y_{re} \mid f_{\theta^*+\Delta\theta}, S_{re}), \tag{8}$$

**(3)** *Avoiding the performance drop caused by unlearning.* Due to the data-driven nature of current machine learning methods, the size deduction of $S$ caused by the removal of unlearning data $S_{ul}$ could potentially harm the performance of $f$ on test data $S_{te}$. Considering that $l_{re}$ is focused on maintaining model performance in training data $S_{re}$, we propose another loss to improve generalization of $f$ in test data $S_{te}$ as

$$\ell_{reg} = d(Y_{re}, Y_{val}) = ||\mathbb{E}(Y_{re}) - \mathbb{E}(Y_{val})||_2 +$$
$$\sum_{i=2}^{k} ||\mathbb{E}(Y_{re} - \mathbb{E}(Y_{re}))^i - \mathbb{E}(Y_{val} - \mathbb{E}(Y_{val}))^i||_2 \tag{9}$$

where $Y_{re}$ and $Y_{val}$ indicate the predictions of $f_{\theta^*+\Delta\theta}$ on $S_{re}$ and validation dataset $S_{val}$, respectively. $d$ indicates the central moment discrepancy function [36] that measures the distribution difference. **(4)** *Avoiding the overfitting caused by unlearning.* Due to the ResNet-like design of Gradient Transformation, our method $\mathcal{U}_\varphi$ may be prone to overfitting $S_{ul}$ w.r.t. its desired label $\widehat{Y_{ul}}$. Although this behavior is desired by unlearning requests, it potentially harms the generalization ability of $f$ on $S_{ul}$, i.e., the $f$ retrained on $S_{re}$ could perform/generalize well on $S_{ul}$ due to its knowledge learned from $S_{re}$. Refer to Appx. D for examples. To avoid this potential adverse effect, a generalization regularization w.r.t. unlearning is defined as

$$\ell_{ulg} = d(\widehat{Y_{ul}}, Y_{ul}^c) \tag{10}$$

where $\widehat{Y_{ul}}$ denotes the desired predictions on $S_{ul}$. $Y_{ul}^c$ is the prediction of $f_{\theta^*+\Delta\theta}$ on the counterpart of $S_{ul}$, which could be any event related to nodes in $S_{ul}$ (e.g., $v_d$ or $v_j$) that does not occur at $t = 6$ (e.g., $S_{ul}^c = \{(add\ edge, (v_d, v_e), t = 6), (add\ edge, (v_j, v_e), t = 6)\}$).

In this paper, our unlearning method *Gradient Transformation* $\mathcal{U}_\varphi$ is trained by the following loss

$$\ell = \ell_{re} + \alpha\ell_{reg} + \beta\ell_{ul} + \gamma\ell_{ulg} \tag{11}$$

**Distinction from previous studies**. **(1)** Our method can be used in a general and post-processing manner to implement DGNN unlearning. It is independent of one specific architecture (e.g., Transformer-based model DyGFormer [35]) and maintains the invariant architecture of the target DGNNs, which overcomes the *model-specific* and *changing architecture* limitations of current methods for (static) graph unlearning. **(2)** For resource concerns, the influence function-based methods ask for $O(n^2)$ space to store the Hessian matrix, where $n$ denotes the parameter size of target models. In contrast, the space requirement of our method is $O(nd_W)$, where $d_W$ indicates the average embedding dimension in Eq. (6) and $d_W << n$. Although this design greatly reduces the resource requirement of our method, it potentially faces a resource bottleneck when handling larger DGNN models in the future, which will be the future work of this paper. **(3)** With the aim of unlearning DGNNs, this paper is motivated to address the limitations of exiting static graph unlearning methods. Although the generality advantage enables its potential in unlearning other models (e.g., models for static graphs or images), this paper focuses on DGNNs for link predictions.

## 6 EXPERIMENTS

### 6.1 Experimental Setup

**Datasets**, **DGNNs**, and **Unlearning Requests**. We evaluated our method on six commonly used datasets for dynamic link prediction [2], including Wikipedia, Reddit, MOOC, LastFM [1], UCI [37], and Enron [38]. We use two state-of-the-art DGNNs, i.e., DyGFormer [35] and GraphMixer [39], as the backbone models in our evaluations due to their outperformance (See Appx. E for more details). For data partition and DGNN training, we followed previous work DyGLib [35]. In this paper, we set $k = 2$ in the moment discrepancy function (see Eq. (9) and (10)). The channel/token dimension (i.e., Eq. (6)) is set to 32. For the loss function (11), we set $\alpha = 1.0$, $\beta = 0.1$, and $\gamma = 0.1$. For the unlearning data, we first sample a fixed number of events as initial events and use their historically observed events (e.g., 621445 events in (LastFM, DyGFormer)) as unlearning data $S_{ul}$. For each (dataset, DGNN) combination, we ran five times on RTX 3090 GPUs to obtain the evaluation results.

**Baselines**. In this paper, we focus on evaluating *post-processing* and *general* unlearning methods for DGNNs. Baseline methods include *retraining*, *fine-tuning*, and *fine-tuning only with unlearning requests*. Also, we use *retraining* from scratch as the gold standard to

**Table 2:** $\text{AUC}(S_{te})$ comparison between our method and baseline methods ($\Delta\text{AUC}(S_{te})$ ↑).

| Methods | Retraining | Fine-tuning-*ul* | Fine-tuning | **Ours** |
|---|---|---|---|---|
| Datasets | DyGFormer | | | |
| Wikipedia | 0.9862 ± 0.0004 | 0.8491 ± 0.1855 | 0.9542 ± 0.0334 | **0.9859 ± 0.0004** |
| UCI | 0.9422 ± 0.0005 | 0.7855 ± 0.2867 | 0.7850 ± 0.2864 | **0.9396 ± 0.0020** |
| Reddit | 0.9885 ± 0.0006 | 0.6459 ± 0.1049 | 0.9479 ± 0.0068 | **0.9902 ± 0.0001** |
| MOOC | 0.7477 ± 0.0289 | 0.5342 ± 0.0986 | 0.7949 ± 0.0275 | **0.8509 ± 0.0032** |
| LastFM | 0.8675 ± 0.0154 | 0.4409 ± 0.0967 | 0.5846 ± 0.0679 | **0.7953 ± 0.0523** |
| Enron | 0.8709 ± 0.0312 | 0.3109 ± 0.1018 | 0.8240 ± 0.0568 | **0.8452 ± 0.0867** |
| Datasets | GraphMixer | | | |
| Wikipedia | 0.9632 ± 0.0017 | 0.9576 ± 0.0032 | 0.9578 ± 0.0031 | **0.9683 ± 0.0016** |
| UCI | 0.9089 ± 0.0052 | 0.9174 ± 0.0072 | **0.9182 ± 0.0068** | 0.9149 ± 0.0069 |
| Reddit | 0.9650 ± 0.0001 | 0.9434 ± 0.0093 | 0.9444 ± 0.0086 | **0.9717 ± 0.0004** |
| MOOC | 0.8257 ± 0.0060 | 0.8179 ± 0.0062 | 0.8179 ± 0.0062 | **0.8367 ± 0.0056** |
| LastFM | 0.7381 ± 0.0020 | 0.7304 ± 0.0011 | 0.7304 ± 0.0011 | **0.7358 ± 0.0014** |
| Enron | 0.8462 ± 0.0009 | **0.8495 ± 0.0035** | **0.8495 ± 0.0035** | 0.8490 ± 0.0036 |

*  In this table, the values indicate the average and variance results of five runs, and the results with largest $\Delta\text{AUC}(S_{te})$ among Fine-tuning-*ul*, Fine-tuning, and our method is highlighted in **bold**.

evaluate unlearning methods. As a typical post-processing method, the fine-tuning method uses the loss $\ell_{re} + \ell_{ul}$ to update the DGNN parameters. To evaluate how overfitting a model will be when only using unlearning requests, we use a variant of fine-tuning that only uses $\ell_{ul}$. Note that current SISA and influence function-based methods are not suitable as baselines here. This is because the former is designed as a pre-processing method, while the resource-intensive issue of the latter invalidates its practicability in the unlearning of DGNNs. Refer to Appx. B.2 for details.

**Metrics**. We use the retrained model as the criterion, and the unlearned model $f_{(\theta^*+\Delta\theta)}$ is expected to perform similarly to or better than $f_{\theta^*_{ul}}$. Specifically, the evaluation of an unlearning method includes three different aspects of metrics, i.e., *model effectiveness*, *unlearning effectiveness*, and *unlearning efficiency*.

**(1)** *Model Effectiveness.* A DGNN $f$ is trained to serve its users in downstream tasks, where its performance is vital during its inference period. In this paper, we use $\Delta\text{AUC}(S_{te}) = \text{AUC}(Y, f_{\theta^*+\Delta\theta}(S_{te})) - \text{AUC}(Y, f_{\theta^*}(S_{te}))$ as the metric, where $S_{te}$ represents the test dataset and AUC indicates the numeric value of the area under the ROC curve (AUC). A higher $\Delta\text{AUC}(S_{te})$ indicates a better unlearning method. Similarly, we also use $\Delta\text{Acc}(S_{re})$ to evaluate unlearning methods w.r.t. model performance on the remaining data $S_{re}$, where Acc indicates the accuracy function.

**(2)** *Unlearning Effectiveness.* According to the data-driven nature of DGNNs, a well-retrained model $f_{\theta^*_{ul}}$ could generalize well on unseen samples (e.g. $S_{ul}$) because it has learned knowledge from $S_{re}$ and the same data patterns potentially exist in both $S_{re}$ and $S_{ul}$. Therefore, this paper uses $|\Delta\text{Acc}(S_{ul})|$ to evaluate the effectiveness of unlearning. A lower value indicates a better unlearning method.

**(3)** *Unlearning Efficiency.* We compare the average time cost $\text{t}_{ave}$ and the speed-up of different methods to evaluate their efficiency.

### 6.2 Model Performance on test data $S_{te}$

As the prediction quality is vital for DGNNs in serving users, we first evaluate their performance on $S_{te}$. Table 2 confirms the outperformance of our method w.r.t. $\Delta\text{AUC}(S_{te})$ ↑. We also observed that: **(1)** Due to the complex Transformer-based design, the retrained DyG-Former outperforms other backbone models in most cases, which

is consistent with previous research [35]. **(2)** Among all unlearning methods, the fine-tuning-*ul* only uses the unlearning loss $\ell_{ul}$ (see Eq. (7)) to obtain the target DGNN parameters, which harms their performance in test data $S_{te}$. For example, with the DyGFormer, the average AUC score on the Enron dataset is only 0.3109, which is extremely below that of retraining, i.e., 0.8709. This observation indicates that it is vital to take into account the remaining training data $S_{re}$ to reduce the performance cost of unlearning methods. **(3)** As an integrated version, the fine-tuning method performs better than the fine-tuning-*ul* method due to the use $\ell_{re} + \ell_{ul}$ as the loss in optimizing the model parameters. However, the performance compromise still exists in most cases. **(4)** Note that even in the cases where our method is not the best, it still has competitive performance. This can be attributed to $\ell_{reg} = d(Y_{re}, Y_{val})$ in the loss function, which helps $f$ generalize well on the test data $S_{te}$.

### 6.3 Model Performance on remaining data $S_{re}$ and unlearning data $S_{ul}$

Table 3 showcases the prediction accuracy comparison among unlearning methods on $S_{re}$ and $S_{ul}$. Specifically, **(1)** although baselines are potentially at the cost of the model performance, our method still generally performs better in the remaining training dataset $S_{re}$ w.r.t. $\text{Acc}(S_{re})$ ↑. For example, for DyGFormer on the Reddit dataset, the $\Delta\text{Acc}$ of the fine-tuning method is −0.0868 while our method still maintains the same level of prediction performance (i.e., $\Delta\text{Acc}$ = +0.0017). **(2)** For the unlearning request $S_{ul}$, a retrained model can still predict well on unseen samples $S_{ul}$ due to its generalization ability. However, as indicated by the underline results, when comparing with the retraining method, we observed that the fine-tuning and fine-tuning-*ul* methods have extreme overfitting to $S_{ul}$. In contrast, our method performs similarly to the retraining method w.r.t. $|\Delta\text{Acc}(S_{ul})|$ ↓, which highlights the outperformance of our method in approximating unlearning.

### 6.4 Unlearning Efficiency

Figure 3 demonstrates the time cost of different methods to implement unlearning, which shows that our method is almost always more efficient than all other baselines. Moreover, we observe that:

**Table 3: Acc comparison between our method and baseline methods.**

| Datasets | Methods | DyGFormer | | GraphMixer | |
|---|---|---|---|---|---|
| | | Acc($S_{re}$)/ΔAcc ↑ | Acc($S_{ul}$)/\|ΔAcc\| ↓ | Acc($S_{re}$)/ΔAcc ↑ | Acc($S_{ul}$)/\|ΔAcc\| ↓ |
| Wikipedia | Retraining | 0.9507 ± 0.0004 | 0.1470 ± 0.0435 | 0.9105 ± 0.0028 | 0.2050 ± 0.0078 |
| | Fine-tuning-*ul* | 0.7541 ± 0.1463 | **0.1574 ± 0.1005** | 0.9112 ± 0.0056 | **0.1666 ± 0.0146** |
| | Fine-tuning | 0.9062 ± 0.0545 | 0.3962 ± 0.0797 | 0.9117 ± 0.0056 | 0.1640 ± 0.0111 |
| | **Ours** | **0.9529 ± 0.0008** | 0.1773 ± 0.0330 | **0.9307 ± 0.0016** | 0.1339 ± 0.0145 |
| UCI | Retraining | 0.8458 ± 0.0008 | 0.2056 ± 0.0097 | 0.8685 ± 0.0003 | 0.1361 ± 0.0080 |
| | Fine-tuning-*ul* | 0.7552 ± 0.1427 | 0.3850 ± 0.3515 | 0.8705 ± 0.0036 | 0.1246 ± 0.0066 |
| | Fine-tuning | 0.7572 ± 0.1437 | 0.4006 ± 0.3408 | 0.8710 ± 0.0034 | 0.1235 ± 0.0080 |
| | **Ours** | **0.8484 ± 0.0020** | **0.2057 ± 0.0028** | **0.8737 ± 0.0019** | **0.1258 ± 0.0055** |
| Reddit | Retraining | 0.9460 ± 0.0012 | 0.0540 ± 0.0033 | 0.9009 ± 0.0003 | 0.0477 ± 0.0028 |
| | Fine-tuning-*ul* | 0.6665 ± 0.0817 | 0.3116 ± 0.2221 | 0.8874 ± 0.0058 | 0.3361 ± 0.0512 |
| | Fine-tuning | 0.8592 ± 0.0104 | 0.2433 ± 0.0727 | 0.8879 ± 0.0060 | 0.3342 ± 0.0516 |
| | **Ours** | **0.9487 ± 0.0006** | **0.0476 ± 0.0014** | **0.9326 ± 0.0005** | **0.0212 ± 0.0021** |
| MOOC | Retraining | 0.8176 ± 0.0124 | 0.0452 ± 0.0389 | 0.8171 ± 0.0033 | 0.1240 ± 0.0068 |
| | Fine-tuning-*ul* | 0.6090 ± 0.1116 | 0.4962 ± 0.3865 | 0.8161 ± 0.0053 | 0.0552 ± 0.0110 |
| | Fine-tuning | 0.7510 ± 0.0252 | 0.4804 ± 0.0595 | 0.8161 ± 0.0053 | 0.0552 ± 0.0110 |
| | **Ours** | **0.7950 ± 0.0036** | **0.0472 ± 0.0164** | **0.8222 ± 0.0026** | **0.0630 ± 0.0130** |
| LastFM | Retraining | 0.8738 ± 0.0049 | 0.3110 ± 0.1099 | 0.6583 ± 0.0006 | 0.2918 ± 0.0075 |
| | Fine-tuning-*ul* | 0.5578 ± 0.0660 | 0.5322 ± 0.2697 | 0.6593 ± 0.0024 | 0.3059 ± 0.0071 |
| | Fine-tuning | 0.7590 ± 0.0163 | 0.9103 ± 0.0232 | 0.6597 ± 0.0024 | **0.2980 ± 0.0030** |
| | **Ours** | **0.8198 ± 0.0224** | **0.2834 ± 0.1714** | **0.6632 ± 0.0015** | 0.3009 ± 0.0080 |
| Enron | Retraining | 0.9321 ± 0.0119 | 0.2101 ± 0.0399 | 0.8007 ± 0.0009 | 0.2866 ± 0.0215 |
| | Fine-tuning-*ul* | 0.5012 ± 0.0007 | 0.9953 ± 0.0059 | 0.8109 ± 0.0015 | **0.2938 ± 0.0569** |
| | Fine-tuning | **0.9015 ± 0.0329** | 0.2759 ± 0.1047 | **0.8110 ± 0.0015** | 0.2771 ± 0.0498 |
| | **Ours** | 0.8226 ± 0.1652 | **0.2324 ± 0.2934** | 0.8100 ± 0.0012 | 0.2944 ± 0.0341 |

\* Acc($S_{re}$) results represent the model performance on remaining training data, and the results (excluding the retraining method) with largest ΔAcc is highlighted in **bold**. Acc($S_{ul}$) results indicate the model performance on unlearning data, and the results with the smallest \|ΔAcc\| are highlighted in **bold**. Underline points out the result where there is an extreme overfitting on $S_{ul}$.

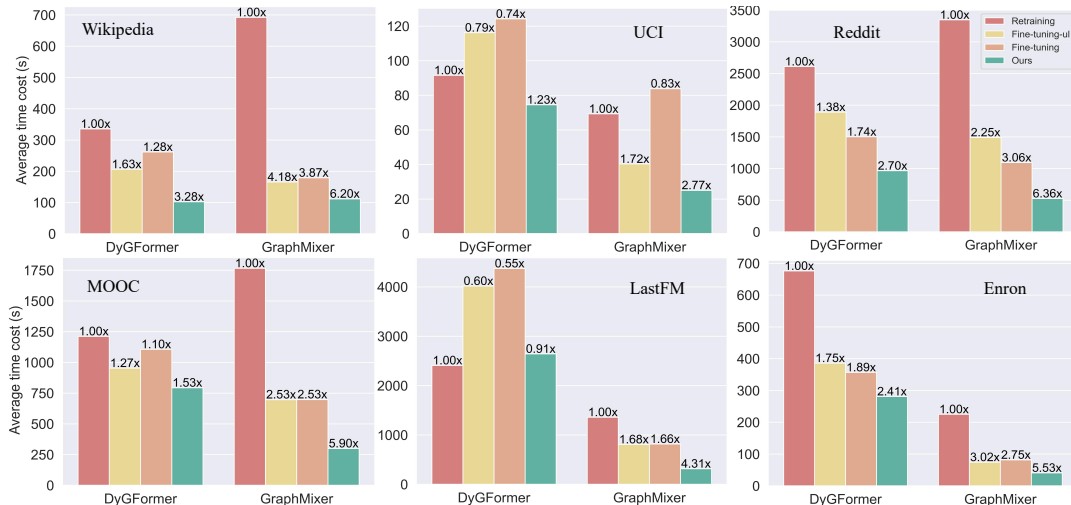

**Figure 3: The time cost comparison between our *Gradient Transformation* method and baseline methods. The numerical values on the bars (e.g., 6.20×) indicate the degree of acceleration relative to the retraining approach.**

**(1)** Due to the use of additional $S_{re}$ in the unlearning process, the fine-tuning method is generally slower than the fine-tuning-*ul* method. **(2)** Although the baselines can make the unlearning process efficient, the overfitting on $S_{ul}$ limits their speed to obtain their optimal solutions, where it is not trivial to navigate between the unlearning effectiveness and model performance.

*Remark.* For most (dataset, DGNN) combinations cases, our method obtained obvious efficiency advantages such as 6.36× speeding up in the case (Reddit, GraphMixer). However, in the (LastFM, DyG-Former) case, the slight slowness can be attributed to the large scale (i.e., 621445 events) of $S_{ul}$ and the trade-off between $S_{ul}$ and $S_{re}$ on model performance. Refer to Appx. F.3 for more details.

**Table 4: Comparison between our method and baseline methods on the CAWN model**

| Datasets | Methods | Acc($S_{re}$)/ΔAcc ↑ | Acc($S_{ul}$)/|ΔAcc| ↓ | AUC($S_{te}$)/ΔAUC ↑ | $t_{ave}(s)$ | Speed-up |
|---|---|---|---|---|---|---|
| Wikipedia | Re-training | 0.9491 ± 0.0006 | 0.1159 ± 0.0048 | 0.9841 ± 0.0002 | 1903.3736 | 1× |
| | Fine-tuning-ul | 0.9485 ± 0.0015 | 0.1191 ± 0.0091 | 0.9820 ± 0.0014 | 526.8222 | 3.61× |
| | Fine-tuning | 0.9485 ± 0.0015 | **0.1133 ± 0.0074** | 0.9814 ± 0.0021 | 568.0133 | 3.35× |
| | **Ours** | **0.9486 ± 0.0007** | 0.1086 ± 0.0046 | **0.9829 ± 0.0001** | **263.2861** | **7.23×** |
| Reddit | Re-training | 0.9429 ± 0.0009 | 0.0411 ± 0.0029 | 0.9890 ± 0.0002 | 7449.4465 | 1× |
| | Fine-tuning-ul | 0.6599 ± 0.0971 | 0.4201 ± 0.2954 | 0.6759 ± 0.1405 | 2500.3047 | 2.98× |
| | Fine-tuning | 0.9041 ± 0.0034 | 0.4559 ± 0.0402 | 0.9640 ± 0.0013 | 3623.2015 | 2.06× |
| | **Ours** | **0.9481 ± 0.0012** | **0.0303 ± 0.0022** | **0.9897 ± 0.0001** | **1329.4738** | **5.60×** |
| Enron | Re-training | 0.9094 ± 0.0018 | 0.1192 ± 0.0061 | 0.8758 ± 0.0094 | 1334.3856 | 1× |
| | Fine-tuning-ul | 0.5399 ± 0.0744 | 0.6818 ± 0.3898 | 0.5970 ± 0.0591 | 537.0440 | 2.48× |
| | Fine-tuning | 0.8600 ± 0.0190 | 0.2534 ± 0.0708 | 0.8403 ± 0.0221 | 612.8398 | 2.18× |
| | **Ours** | **0.9162 ± 0.0007** | **0.1120 ± 0.0026** | **0.9024 ± 0.0007** | **291.0586** | **4.58×** |

* Acc($S_{re}$) results represent the model performance on remaining training data, and the results (excluding the retraining method) with largest ΔAcc is highlighted in **bold**. Acc($S_{ul}$) results indicates the model performance on unlearning data, and the results with smallest |ΔAcc| are highlighted in **bold**. Underline points out the result where there is extremely overfitting on unlearning requests. AUC($S_{te}$) results show the model performance on test data, and the the results (excluding the retraining method) with largest ΔAUC is highlighted in **bold**.

**Table 5: Comparison between our method and the retraining on the Wikipedia dataset when dealing future unlearning requests**

| | | Acc($S_{re}$) | Acc($S_{ul}$) | AUC($S_{te}$) | $t_{ave}$(s) | Speed-up |
|---|---|---|---|---|---|---|
| DyGFormer | Retraining | 0.9525 ± 0.0009 | 0.0514 ± 0.0035 | 0.9869 ± 0.0005 | 506.4784 | 1× |
| | **Ours** | 0.9569 ± 0.0065 | 0.0722 ± 0.0099 | 0.9853 ± 0.0004 | 16.2926 | 31.09 × |
| GraphMixer | Retraining | 0.9077 ± 0.0056 | 0.0977 ± 0.0079 | 0.9610 ± 0.0031 | 500.6417 | 1× |
| | **Ours** | 0.9278 ± 0.0119 | 0.0803 ± 0.0296 | 0.9600 ± 0.0137 | 19.9474 | 25.10 × |
| CAWN | Retraining | 0.9495 ± 0.0007 | 0.0749 ± 0.0013 | 0.9837 ± 0.0002 | 1594.1584 | 1 × |
| | **Ours** | 0.9570 ± 0.0047 | 0.0803 ± 0.0199 | 0.9830 ± 0.0002 | 48.9181 | 32.59 × |

**Table 6: Ablation study on loss with (DyGFormer, LastFM).**

| Methods | Acc($S_{re}$) | Acc($S_{ul}$) | AUC($S_{te}$) |
|---|---|---|---|
| Re-training | 0.8738 | 0.3110 | 0.8675 |
| Fine-tuning-*ul* | 0.5578 | 0.5322 | 0.4409 |
| Fine-tuning | 0.7590 | 0.9103 | 0.5846 |
| **Ours**-*re-ul* | 0.8239 | 0.2758 | 0.7988 |
| **Ours**-*re-ul-reg* | **0.8241** | 0.2790 | **0.8006** |
| **Ours**-*re-ul-ulg* | 0.8170 | 0.2759 | 0.7931 |
| **Ours**-*full* | 0.8198 | **0.2834** | 0.7953 |

* This table shows the average evaluation results. Following the metric in Tables 2 and 3, the best method is highlighted in **bold**.

## 6.5 Other Evaluations

**Method Generality**. We further verify that our method is independent of the DGNN architecture by evaluating it on the CAWN model [40], whose random walking-based architecture is different from that of DyGFormer and GraphMixer. Consistent with Tables 2 and 3, the results in Table 4 confirm that our method is *effective*, *efficient*, and *general* for the unlearning of DGNNs.

**Prediction Similarity Comparison**. Note that one of the unlearning goals is to eliminate the influence of $S_{ul}$ when applying the target model to the test data $S_{te}$ [41]. Thus, we compared prediction similarity to evaluate the extent of unlearning of our method. Evaluation results indicate that our method achieves an average unlearning rate 81.33% on the test data (See Appx. F.1 for details).

**Ablation study on loss function.** We evaluated the contribution of different loss items in Eq. (11). Compared with the basic version (i.e., **Ours**-*re-ul*), additionally introducing the utility generalization loss $\ell_{reg}$ (i.e., **Ours**-*re-ul-reg*) improves the model performance on $S_{re}$ and $S_{te}$, while using $\ell_{ulg}$ (i.e., **Ours**-*re-ul-ulg*) decreases the model utility on them. In contrast, the comprehensive version (i.e., **Ours**-*full*) obtains the best unlearning results on $S_{ul}$.

**Unlearning without training**. The learning nature of our method brings potential benefits when handling future unlearning requests. Using the gradient of new unlearning requests as input, our method could directly output the desired parameter update. In this paper, we use the sampled initial events (refer to Section 6.1) as future unlearning requests to evaluate our method. Table 5 shows that our method obtains almost the same unlearning results as the re-training method. For example, besides performance increments, the largest |ΔAcc($S_{ul}$)| is only 0.0208 while the speed-up is at least 25×. However, this benefit of our method could be limited due to the complex relationships between $S_{re}$ and $S_{ul}$ and the lack of ground truth $\Delta\theta$ in the training of our method. In cases where the benefit is limited, model developers can re-run our method in Figure 2 to conduct future unlearning.

## 7 CONCLUSION

In this paper, we study dynamic graph unlearning and propose a method called Gradient Transformation, which is *effective*, *efficient*, *general*, and can be used in a *post-processing* manner. Empirical evaluations on real-world datasets confirm the effectiveness and efficiency outperformance of our method, and we also demonstrate its potential advantages in handling future unlearning requests. In the future, we will study the causal relationships between events from the unlearning view, while also delving into the intricate interplay between the remaining data and the unlearning data.

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

# A  CONCEPTS

**Discrete-time Dynamic Graphs (DTDG).** Given a benign sequence of static graphs with time length $T = t - 1$, a discrete-time dynamic graph is denoted as $S = \{G_1, G_2, \ldots, G_{t-1}\}$, where $G_k = \{\mathcal{V}_k, \mathcal{E}_k\}$ denotes the $k$-th snapshot of a dynamic graph. In the form of $G_k = (\mathbf{A}_k, \mathbf{X}_k)$, $\mathbf{A}_{k;i,j} = 1$ if there is a link from $v_i$ pointing to $v_j$ in the $k$-th snapshot, otherwise $\mathbf{A}_{k;i,j} = 0$; $\mathbf{X}_{k;i}$ represents the node feature of $v_i$ in $G_k$, and $\mathbf{X}_{k;i,j}$ indicates its $j$-th feature value. Thus, a discrete-time dynamic graph can also be depicted as $S = \{(\mathbf{A}_1, \mathbf{X}_1), (\mathbf{A}_2, \mathbf{X}_2), \ldots, (\mathbf{A}_{t-1}, \mathbf{X}_{t-1})\}$.

# B  RELATED WORK

## B.1  Dynamic Graph Neural Networks

The dynamic graph is a powerful data structure that depicts both spatial interactions and temporal changes in practical data from various real-world applications (e.g., traffic prediction [42]). Dynamic graph neural networks (DGNNs) are proposed to learn the complex spatial-temporal patterns in these data [31, 33, 43]. Next, we will present some representative methods to introduce how dynamic graph neural networks learn from dynamic graphs. Depending on the type of dynamic graph (as shown in Section 3), current methods can be categorized into DGNNs for discrete-time and continuous-time dynamic graphs [35].

**(1)** For discrete-time methods, they generally employ a GNN model for static graphs to learn spatial representations and an additional module (e.g., RNN [44]) to capture the temporal changes of the same node in different static snapshot graphs. In this paper, we focus on DGNNs designed for continuous-time dynamic graphs. Unlike discrete-time dynamic graphs, this type of data naturally records dynamic changes (i.e., fine-grained order of different changes) without determining the time interval among the snapshots in discrete-time dynamic graphs [45]

**(2)** Some typical continuous-time methods are RNN-based models (e.g., JODIE [1]), temporal point process models (e.g., DyRep [46]), time embedding-based models (e.g., TGAT [45], TGN [47]), and temporal random walk methods [33, 40].

In this paper, our evaluations focus on the DyGFormer and GraphMixer models for two reasons. (1) According to a recent study [35], these two methods have obvious outperformance over previous methods (e.g., JODIE [1], DyRep [46], TGAT [45], TGN [47]), allowing them to be preferentially deployed in real-world applications. (2) These methods have different architectures, which helps in comprehensively evaluating our methods. The DyGFormer [35] is designed using the famous Transformer module, and the GraphMixer [39] is an MLP-based framework. Moreover, we also evaluated our method on the CAWN model [40], which is an approach based on the temporal random walk.

## B.2  Unlearning Methods

**Retraining**. Intuitively, for the target model to be unlearned, deleting the unlearning data from the original training data and retraining from scratch will directly meet the requirement from the perspective of unlearning. However, retraining from scratch comes at the cost of huge time and computational resources, given large-scale training data or complex model architectures. To this end,

various methods have been proposed to satisfy the *efficiency* requirement of unlearning. Current methods designed for GNNs focus on static graph unlearning, including SISA methods, influence function methods, and other approaches.

**SISA Methods**. Referring to "*Sharded, Isolated, Sliced, and Aggregated*", the SISA method represents a type of ensemble learning method and is not sensitive to the architecture of target models (i.e., model-agnostic). Specifically, SISA first divides the original training data $\mathcal{D}_o$ into $k$ different and disjoint shard datasets, i.e., $\mathcal{D}_o^1, \ldots, \mathcal{D}_o^k$, which are used to train $k$ different submodels $f^1, \ldots, f^k$ separately. To obtain the final prediction on a sample $v_i$, SISA aggregates $f^1(v_i), \ldots, f^k(v_i)$ together to obtain a global prediction. Upon receiving the unlearning request for a sample $v_j$, SISA first removes $v_j$ from the shard $\mathcal{D}_o^j$ that includes it and only retrains $f^j$ to obtain the updated model, which significantly reduces the time cost compared with retraining the whole model from scratch on $\mathcal{D}_o \setminus v_j$ (i.e., the dataset with removing $v_j$).

*Typical methods*. The SISA method divides the training dataset into several subsets and trains submodels on them, followed by assembling these submodels to serve users. In the context of graph unlearning, it is not trivial to directly split a whole graph into several subgraphs, as imbalanced partition (e.g., imbalance of node class) potentially leads to decreased model performance. To this end, GraphEraser [16] and RecEraser [18] propose balanced graph partition frameworks and learning-based aggregation methods. Unlike the transductive setting of GraphEraser, Wang *et al.* [22] propose a method called GUIDE in the inductive learning setting, which takes the fair and balanced graph partitioning into consideration.

*Limitations*. The weakness of SISA methods mainly includes the following two aspects.

**(1)** *Efficacy Issue on A Group of Unlearning Requests.* The SISA method faces the efficiency issue when dealing with a group/batch of unlearning requests. Note that the efficiency of SISA methods in facilitating unlearning stems from the fact that retraining a single submodel is more efficient than retraining the entire model that was trained on the whole dataset, which is suitable for implementing unlearning of a single sample. However, the SISA method will have to retrain all submodels when a group of unlearning requests binds to all shards, limiting its unlearning efficiency capacity.

**(2)** *Reliance on Pre-processing*. Note that SISA methods can only be used in the initial phase of model development. Once the machine learning models are deployed, the current SISA methodology cannot implement unlearning on them in a post-processing manner.

**Influence Function based Methods**. Current research on the influence function studies how a training sample impacts the learning of a machine learning model [19, 20]. Generally, given a model $f$, its optimal parameter is obtained by $\theta^* = \text{argmin}_\theta \sum_{v \in \mathcal{D}_o} \ell(f_\theta \mid \mathcal{D}_o)$, where $\ell$ indicates a convex and twice-differential loss function. For the unlearning request on a training sample $v_j$, the desired model parameter $\theta_{ul}^*$ is defined as $\theta_{ul}^* = \text{argmin}_\theta \sum_{v \in \mathcal{D}_o \setminus v_j} \ell(f_\theta \mid \mathcal{D}_o \setminus v_j)$. Without retraining $f$, current influence function-based methods are designed to estimate the parameter change $\Delta\theta$ and use it to approximate $\theta_{ul}^*$ (i.e., $\theta_{ul}^* \approx \theta^* + \Delta\theta$). The estimation can be obtained by $\Delta\theta = \mathbf{H}_{\theta^*} \nabla_{\theta^*} \ell(f_\theta \mid v_j)$, where $\mathbf{H}_{\theta^*}^{-1} = \sum_{v \in \mathcal{D}_o} \nabla_{\theta^*}^2 \ell(f_\theta \mid \mathcal{D}_o)$ denotes the Hessian matrix $\mathbf{H}$.

*Typical methods.* Current research on the influence function studies how a training sample affects the learning of a machine learning model [19, 20]. In the context of graph unlearning, the influence function is more complex since the edges between nodes break the independent and identically distributed assumption of training samples in the above formulations. Therefore, the multi-hop neighbors of an unlearning node or nodes involved in an unlearning edge have been considered to correct the above estimation of $\Delta\theta$, and more details can be found in recent works called Certified Graph Unlearning (CGU) [23], GIF [24], CEU [25], and an unlearning method based on Graph Scattering Transform (GST) [48]. Note that these methods generally rely on the static graph structure to determine the scope of nodes that need to be involved in the final influence function, which cannot be directly adapted to dynamic graphs.

*Limitations.* The weakness of influence function-based methods mainly includes the following two aspects.

**(1)** *Model-specific design.* Current estimations of $\Delta\theta$ in graph unlearning are generally established on simple GNN models (e.g., linear GCN model in CEU [25]), whose estimation accuracy is potentially limited due to the complexity and diversity of current state-of-the-art GNN architectures (e.g., Graph Transformer Networks [28]). Furthermore, these methods are not strictly model-agnostic, since they need access to the architecture of target models (e.g., layer information) to determine the final influence function (e.g., GIF [24]), that is, they are sensitive to GNN layers.

**(2)** *Resource intensive.* Although influence function-based methods can be used in a post-processing manner, the calculation of the inverse of a Hessian matrix (i.e., $\mathbf{H}_{\theta*}^{-1}$) generally has a high time complexity and memory cost when the model parameter size is large. For example, for a DyGFormer model [35] with single precision floating point format parameters, it requires **4.298 TB** space to store the Hessian matrix when the parameter size is 4.146 MB (i.e., the model on Wikipedia dataset). This resource issue severely limits the practicability of influence function-based methods in the unlearning of dynamic graph neural networks.

**Others.** Recently, the PROJECTOR method proposes mapping the parameter of linear GNNs to a subspace that is irrelevant to the nodes to be forgotten [21]. However, it is a model-specific method and cannot be applied to the unlearning of other non-linear GNNs. Unlike existing model-specific methods, GNNDelete [26] introduces an **architecture modification** strategy, where an additional trainable layer is added to each layer of the target GNN. Although this method is model-agnostic, the requirement of additional architecture space reduces its practicability in scenarios where the deployment space is limited (e.g., edge computing devices [29]). Another model-agnostic method called GraphGuard [12] uses **fine-tuning** to make the target GNN forget the unlearning samples, where both the remaining and unlearning data are involved in the loss function to fine-tune the model parameter. However, the fine-tuning method is prone to overfitting unlearning samples [30], which potentially harms the model performance.

## C   UNLEARNING REQUESTS OF DGNNS

Any data change needs in the training data $S$ could potentially raise unlearning requests. Besides the unlearning requests in Section 4, other unlearning needs also exist in real-world applications for various reasons. For example, users may expect to unlearn some specific features (e.g., forgetting the gender and age information for system fairness) or labels (e.g., out-of-date tags of interest) [41]. **Interaction between $S_{ul}$ and $S_{re}$.** In AI systems for general data (e.g., image or tabular data), $S_{ul}$ and $S_{re}$ are independent of each other, since the samples in the training dataset are *independent and identically distributed* (IID). However, for (static) graph data, $S_{ul}$ and $S_{re}$ can potentially interact with each other [17]. For example, given an unlearning node $v_i \in S_{ul}$, its connected neighbor nodes may belong to the remaining dataset $S_{re}$. Due to the message passing mechanism and stacking layer operation in most GNNs, existing studies have proposed to consider multi-hop neighbors when unlearning is performed for GNNs [25].

As shown in Figure 4, DGNNs are designed to learn both spatial and temporal information in dynamic graphs, making the interaction between $S_{ul}$ and $S_{re}$ in dynamic graphs more complex than that in static graphs. For example, in a dynamic graph neural network $f$ for CTDG data, the embedding of a node at time $t$ is obtained by taking into account its spatial and historical neighbor nodes. In the context of unlearning a DGNN $f_{\theta*}$, i.e., obtaining the desired parameter $\theta_{ul}^*$ with an unlearning method $\mathcal{U}$, the complex interaction between $S_{ul}$ and $S_{re}$ includes:

- *Changed predictions on invariant representations.* Before and after removing $S_{ul}$ from $S$, an event $o_i$ in $S_{ul}$ may have observed the same series of events that remained in $S_{re}$ before its occurrence time, where spatial-temporal subgraphs are prone to generating almost invariant node embedding. However, the unlearning method $\mathcal{U}$ expects $f$ to make different predictions (e.g., $(v_d, v_j)$ at $t = 6$).
- *Invariant predictions on changed representations.* Due to the removal of $S_{ul}$, the spatial-temporal neighbors of an event $o_j \in S_{re}$ are potentially different (e.g., $v_j$ at $t = 8$), while the unlearning method $\mathcal{U}$ expects $f$ to make invariant predictions (e.g., there is an edge $(v_i, v_j)$ at $t = 8$). Note that it is also not trivial to identify exactly the events in $S_{re}$ that are influenced by the unlearning request $S_{ul}$.

## D   EXAMPLES OF AVOIDING OVERFITTING UNLEARNING DATA

As shown in Figure 5, in the remaining training data $S_{re}$, there is an edge event between $v_a$ and $v_i$ at time $t = 5/7$ because they share a common neighbor node at the last time point (i.e., there are edges $(v_a, v_c)$ and $(v_i, v_c)$ at $t = 4/6$). Although samples with the same pattern have been included in $S_{ul}$ (e.g., the edge $(v_j, v_d)$ at $t = 6$ because they share the same neighbor $v_f$ in $t = 5$), a well-retrained model $f$ can generalize well on $S_{ul}$ when a lot of samples with the same patterns have been kept in the remaining data $S_{re}$. Therefore, focusing on 100% unlearning of $S_{ul}$ (e.g., there is no edge prediction in $S_{ul}$ with 100% accuracy) potentially harms the performance of DGNNs on $S_{re}$.

The example in Figure 5 and the practical evaluation results on $S_{ul}$ by retraining (i.e., Tables 3 and 4) support us in using the results from the retraining method as the only gold standard to evaluate other unlearning methods. Our efforts to alleviate overfitting $S_{ul}$ also include setting $\beta = 0.1$ in the loss function (11).

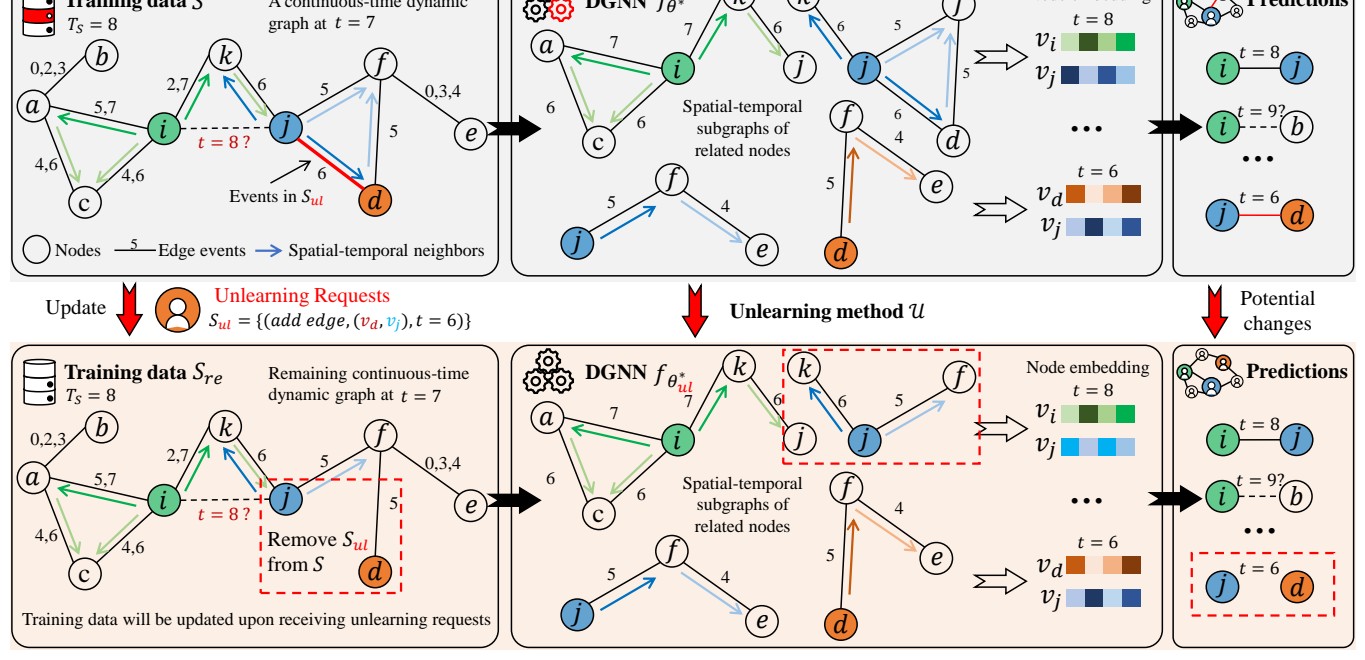

**Figure 4: An overview of the complex interaction between $S_{re}$ and $S_{ul}$. (1) In the upper half, given a dynamic graph $S$ (maximum event time $T_S = 8$) and a DGNN $f$, we use an algorithm $\mathcal{A}_f$ to obtain the optimal $f_{\theta^*}$, which can make accurate predictions on both training ($t \leq T_S$) and test ($t > T_S$) data. (2) Green/blue/brown arrows indicate the spatial and temporal neighbors of node $v_i/v_j/v_d$ in the last 2 historical time points. As shown in the middle column, DGNNs generally use the derived spatial-temporal subgraphs to obtain the node embedding. By combining the embedding of two nodes, $f$ can predict whether there is an edge between them at specific time points. (3) In the lower half, upon receiving the unlearning request $S_{ul}$, an unlearning method aims to approximate the parameter obtained from retraining $f$ with $S_{ul}$. (4) The dashed box on the left indicates the change in training data. The middle dashed box identifies the changed spatial-temporal subgraph of node $v_j$, potentially resulting in changed embedding at time $t = 8$. The right dashed box highlights the desired prediction change from the perspective of unlearning.**

**Table 7: Statistics of the datasets in this paper.**

| Datasets | Domains | #Nodes | #Links | Bipartite | Duration | Unique Steps | Time Granularity |
|---|---|---|---|---|---|---|---|
| Wikipedia | Social | 9,227 | 157,474 | True | 1 month | 152,757 | Unix timestamps |
| UCI | Social | 1,899 | 59,835 | False | 196 days | 58,911 | Unix timestamps |
| Reddit | Social | 10,984 | 672,447 | True | 1 month | 669,065 | Unix timestamps |
| Enron | Social | 184 | 125,235 | False | 3 years | 22,632 | Unix timestamps |
| MOOC | Interaction | 7,144 | 411,749 | True | 17 months | 345,600 | Unix timestamps |
| LastFM | Interaction | 1,980 | 1, 293, 103 | True | 1 month | 1, 283, 614 | Unix timestamps |

# E  DATASETS AND BACKBONE DGNNS

**Datasets**. The six datasets in this paper are commonly used in the current study of dynamic graph neural networks [2, 35], and these datasets can be publicly accessed at the zenodo library. Table 7 presents the basic statistics of these datasets.

**DGNNs**. In this paper, we focus on the following DGNN methods, which have different types of architecture, to evaluate the performance of unlearning methods.

- **DyGFormer**. Motivated by the fact that most existing methods overlook inter-node correlations within interactions, a method

called DyGFormer [35] proposes a transformer-based architecture, which achieves the SOTA performance on dynamic graph tasks like link prediction and node classification.

- **GraphMixer**. In GraphMixer [39], a simple MLP-mixer architecture is used to achieve faster convergence and better generalization performance, excluding complex modules such as recurrent neural networks and self-attention mechanisms, which are employed as de facto techniques to learn spatial-temporal information in dynamic graphs.

- **CAWN**. To obtain node embeddings, CAWN [40] samples random walks for each node and uses the anonymous identity to

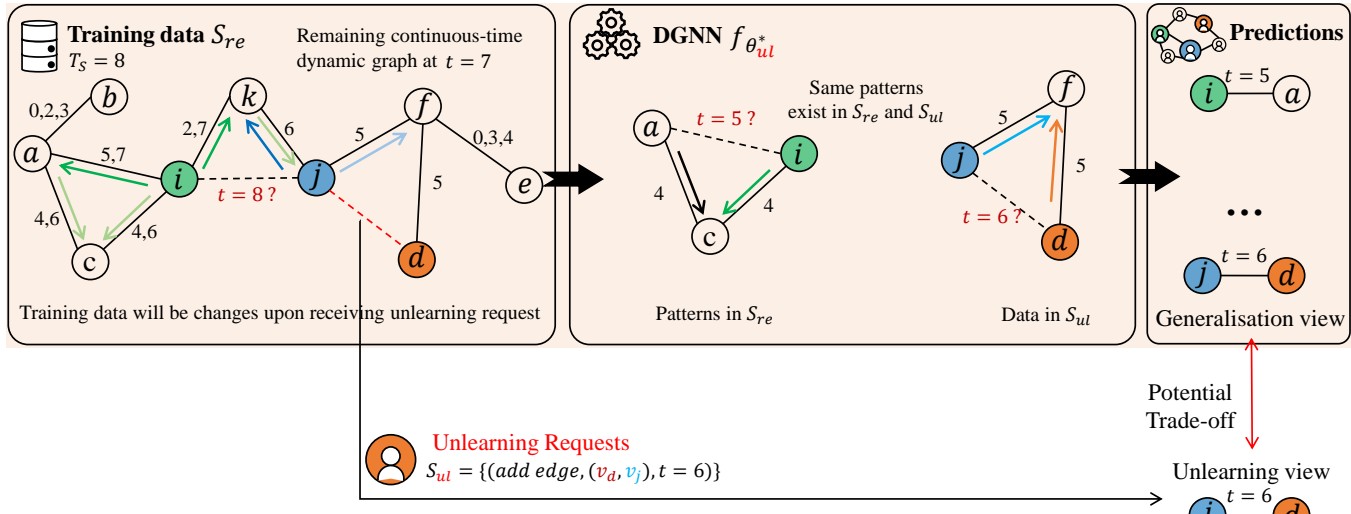

**Figure 5: An illustration of the potential trade-off between model generalization and unlearning requests on the training dataset. See Appx. D for more details.**

denote the nodes in these walks, which helps capture and extract multiple causal relationships in dynamic graphs.

# F  ADDITIONAL EVALUATIONS
## F.1  Original Model vs Unlearned Model Classification

As far as we know, there are no inference methods designed for dynamic graph neural networks, which can infer if an edge is in the training data of the model for link prediction. To evaluate the degree of unlearning, we compare the prediction similarity between the models obtained by our method and the retrained/original model, based on which we assign a class label to the prediction results derived from our method.

According to the motivation of machine unlearning [15], **unlearning methods aim to remove the influence of the unlearning data and make the target model forget the knowledge learned from these data, which can be used to serve users during the inference time of models**. Therefore, we consider the predictions from the test data $Y_{te}^{(our)}, Y_{te}^{(ori)}, Y_{te}^{(ret)}$ as inputs of Eq. (12) to evaluate the unlearning effectiveness from the perspective of ultimate unlearning goal (i.e., $f_{\theta^*+\Delta\theta}(S_{te}) = f_{\theta_{ul}^*}(S_{te})$). Here, we compare the predictions with the test data, on which the unlearned model will be employed to serve users.

**Classification Method**. Assume that the predictions obtained from the original model, the re-trained model, and the model using our method are denoted as $Y^{(ori)}, Y^{(ret)}$, and $Y^{(our)}$, respectively. If $Y^{(our)}$ is regarded as coming from the original model $f_{\theta^*}$, the assigned class label C for $Y^{(our)}$ will be $C(Y^{(our)}) = C(Y^{(ori)}) = C_{\theta^*}$; otherwise, $C(Y^{(our)}) = C(Y^{(ret)}) = C_{\theta_{ul}^*}$. Specifically, **(1)** we obtain the prediction similarity by calculating the percentage of the same predictions, that is, $\mathrm{Acc}(Y^{(our)}, Y^{(ori)})$ and $\mathrm{Acc}(Y^{(our)}, Y^{(ret)})$. **(2)** We treat $Y^{(ori)}$ and $Y^{(ret)}$ as the class center, and use the 1-nearest

neighbor method to categorize $Y^{(our)}$ into the original or unlearned model class. Therefore, the label of $Y^{(our)}$ is obtained by

$$
C(Y^{(our)}) = \begin{cases} C_{\theta_{ul}^*}, & if\ \mathrm{Acc}(Y^{(our)}, Y^{(ret)}) > \mathrm{Acc}(Y^{(our)}, Y^{(ori)}) \\ C_{\theta^*}, & if\ \mathrm{Acc}(Y^{(our)}, Y^{(ret)}) \le \mathrm{Acc}(Y^{(our)}, Y^{(ori)}) \end{cases}
$$
(12)

**Classification Results**. Our approach successfully generates an unlearned model with an average likelihood of 81.33%, based on predictions from test data across all (dataset, DGNN) combinations, where each case is executed five times. These results further validate the effectiveness of our method in conducting an approximate unlearning of DGNNs.

## F.2  Evaluations on future unlearning requests

As shown in Table 5, we compared our method with the retraining method in the Wikipedia dataset when implementing future unlearning requests, which indicates the additional benefits of our method. Due to the training of our method in the previous unlearning process, the evaluation results suggest that it can potentially be used to directly infer the desired parameter update w.r.t. future unlearning requests. Table 5 shows that, in response to future unlearning requests, our approach can produce an unlearned model with a prediction accuracy comparable to the retraining method across the remaining, unlearning, and test data. Importantly, for such unlearning requests, our method can achieve an impressive speed increase, ranging from 25.10× to 32.50×.

## F.3  Evaluations on unlearning efficiency

Figure 3 demonstrates the efficiency advantage of our method. For most (dataset, DGNN) combination cases, our method obtained obvious efficiency advantages such as 6.36× speeding up in the case (Reddit, GraphMixer). However, in the (LastFM, DyGFormer) case, the slight slowness can be attributed to the following reasons.

- The time cost increases when the amount of unlearning samples increases. Note that one typical advantage of our method is that it can deal with a batch of unlearning requests at once. After checking the code, we found that almost half of the training events (48%, unlearn 621445 events, while the total event number is 1293103) are used to compare the performance of different unlearning methods in this case. Compared with only limited unlearning samples (for example, one unlearning sample at a time [16]), the retraining method has fewer training dataset (i.e., the remaining data) when the amount of unlearning sample is larger.

- There is a potential trade-off between the unlearning samples and the remaining samples (as shown in Figure 5). Compared to the retraining method, which only focuses on improving model performance in 52% of the remaining data, other unlearning methods navigating the balance between remaining data and unlearning data generally have a slower rate of convergence, especially when the size difference between remaining and unlearning data is small.

Note that this harsh case (i.e., almost half of the training samples need to be unlearned) is not common in practical scenarios where only a limited ratio of training samples need to be unlearned. As shown in Figure 3, our method has an obvious efficiency advantage in most cases when unlearning a group of requests.

Received 20 February 2007; revised 12 March 2009; accepted 5 June 2009

