# OpenReview forum: "Dynamic Graph Unlearning: A General and Efficient Post-Processing Method via Gradient Transformation"
_ACM.org/TheWebConf/2025/Conference — WWW 2025 Oral_

### Official Review · Reviewer_wrki · 2024-11-17

**Novelty:** 6
**Technical Quality:** 6

**Review:**

The paper proposes a method for "unlearning" in dynamic graph neural networks (DGNNs), addressing current privacy protection and AI governance regulations, particularly the "right to be forgotten" under GDPR. It introduces a gradient transformation-based method for dynamic graph unlearning, which allows efficient parameter updates through post-processing without altering the model architecture. Evaluations on six real-world datasets demonstrate the effectiveness and efficiency of the proposed method, showing significant performance improvements over existing baselines. However, the paper has the following issues:
1.Methods such as SISA and influence function-based approaches are not directly compared in experiments; their limitations are only theoretically discussed.
2.Experiments are conducted on only two DGNN models (DyGFormer and GraphMixer), without covering other common models in the field (e.g., TGAT, JODIE). It is recommended to test on lightweight or differently structured DGNNs to verify the generality of the method.
3.No performance metrics such as memory usage, training time, or generalization ability are evaluated.
4.Although the paper mentions the GDPR's "right to be forgotten," it lacks a thorough discussion of practical issues in real-world applications. This includes risks of privacy breaches during deployment, the potential impact of large-scale unlearning on data-driven industries (e.g., recommendation systems), and possible resource bottlenecks. It is recommended to include discussions on future optimization directions.
5.The most significant issue is the weak connection with Web applications. The authors briefly describe the potential applicability of the method in Web scenarios in the introduction, but this is insufficient for a general-purpose method. The method, as presented, appears more suited for use in most dynamic graph neural networks. The paper provides minimal descriptions of Web-related aspects, making it more appropriate for an artificial intelligence conference.

**Questions:**

1.In high-frequency dynamic updates (e.g., real-time social networks), will the gradient transformation method encounter performance bottlenecks or cumulative biases?
2.Could the unlearning process lead to overfitting or a decline in generalization ability? For example, how can the method ensure optimal results on unlearning requests without negatively affecting the generalization performance on retained data?
3.To what extent can gradient transformation replace actual retraining? Is there any theoretical proof or lower bound analysis to guarantee that the parameter updates from this method do not significantly deviate from retraining results?

**Reviewer Confidence:**

2: The reviewer is willing to defend the evaluation, but it is likely that the reviewer did not understand parts of the paper

**Scope:**

1: The work is irrelevant to the Web

---

### Official Review · Reviewer_L9ja · 2024-11-23

**Novelty:** 6
**Technical Quality:** 4

**Review:**

### Summary

This paper addresses the challenge of implementing machine unlearning in Dynamic Graph Neural Networks (DGNNs). The authors highlight that existing unlearning methods for static graphs are inadequate for dynamic graphs due to their inability to handle dynamic elements and other limitations such as reliance on pre-processing or model-specific designs. To overcome these challenges, the authors propose a post-processing solution called Gradient Transformation, which directly maps unlearning requests to parameter updates using a specialized architecture and loss function. Extensive experiments on six real-world datasets and with state-of-the-art DGNN architectures demonstrate that the proposed method achieves unlearning effectively, and maintains or even improves model utility.

### Pros

1. **Novel problem formulation**: First work to formally address unlearning in dynamic graphs, which has important practical applications.
2. **General and practical solution**: The proposed method works as a post-processing step and is model-agnostic, making it widely applicable.
3. **Good writing and organization**: This paper is well written and clearly organized.

### Cons

1. **Limited discussion of privacy guarantees**: While privacy motivation is mentioned, formal privacy guarantees are not analyzed, which is vital for the practical use of the proposed unlearning method.
2. **Missing some important experiments**: (1) It would be valuable to see how the proposed method performs with different sizes of unlearning requests beyond the current setup.; (2) While the authors justify not including certain baselines, it would be better to compare the proposed method with adapted versions of existing unlearning methods; (3) The method introduces additional hyperparameters ($\alpha$, $\beta$, and $\gamma$) in the loss function, but there is no analysis on how sensitive the unlearning performance is to these parameters or guidelines on how to set them.

**Questions:**

Besides the major concerns outlined above, the reviewer has a few additional questions or comments. It would be great if the authors could also address them.

1. In most cases, the proposed unlearning method can even achieve higher accuracy than the retraining baseline on the remaining data. How is that possible and why is it the case?
2. The paper would benefit from a more detailed discussion of the trade-offs between unlearning effectiveness (accuracy on unlearning data) and model utility (accuracy on remaining data).

**Reviewer Confidence:**

3: The reviewer is confident but not certain that the evaluation is correct

**Scope:**

4: The work is relevant to the Web and to the track, and is of broad interest to the community

---

### Official Review · Reviewer_KXMb · 2024-11-24

**Novelty:** 5
**Technical Quality:** 5

**Review:**

1. Summary
This paper proposes a novel method named "Gradient Transformation" for data privacy protection in dynamic graph neural networks (DGNNs). For the first time, "Dynamic Graph Unlearning" is applied to the context of continuous-time dynamic graphs to address privacy unlearning needs. Through theoretical modeling and experimental validation, gradient transformation demonstrates significant advantages over existing methods in terms of performance, such as model utility and unlearning efficiency. Notably, results on six real-world datasets show up to a 32.59x speed improvement while maintaining comparable accuracy.

2. Strengths
1）Innovation:
First attempt at dynamic graph unlearning: This study is the first to define and implement unlearning requirements in the context of DGNNs, showcasing theoretical and practical innovation.
Gradient transformation method: A framework based on gradient post-processing is introduced, demonstrating architecture independence and post-processing capabilities, making it applicable to complex scenarios that existing methods cannot handle.
2）Performance Improvement:
Efficient unlearning: Experiments show that the gradient transformation method reduces time consumption by 6.2x to 32.59x compared to retraining.
Robustness: The method achieves predictive performance comparable to or better than full retraining across multiple datasets.
Low resource requirements: The method significantly reduces memory usage compared to influence function-based approaches that rely on Hessian matrices.
3）Theoretical and Experimental Support:
The method rigorously defines the design objectives of unlearning requests and validates its optimization through mathematical modeling.
The experiments span multiple mainstream DGNNs and real-world datasets, lending strong credibility to the results.

3. Weaknesses and Questions
1）Unclear definition of terms and objectives:
The definition of "Dynamic Graph Unlearning" and the specific goals of unlearning requests are vague, especially regarding the distinction from static graph unlearning. This might hinder readers' understanding of the unique challenges and contributions of dynamic graph unlearning.
Key concepts, such as the theoretical foundation of "gradient transformation" and its applicability in various scenarios, lack sufficient elaboration, leading to potential confusion.
2）Clarification of Methodological Limitations:
The paper attributes the limitations of prior methods to their reliance on preprocessing, but this is not entirely accurate. The main issue is that these methods cannot support the dynamic update requirements of DGNNs and are only applicable to static scenarios during the early stages of development, lacking support for subsequent dynamic unlearning tasks. The authors are advised to revisit and accurately describe the true limitations of previous methods to enhance the rigor of the paper.
3）Insufficient discussion on practical deployment feasibility:
While the method demonstrates advantages in resource efficiency and time performance, potential challenges in practical deployment, such as adaptability in distributed environments and computational requirements on edge devices, are not sufficiently discussed. The impact of computational bottlenecks and latency in high-frequency unlearning request scenarios requires further analysis.
4）Lack of Explanation for Hyperparameter Settings:
The hyperparameter settings in Equation (11) lack detailed justification, and it is unclear why the current values were chosen and how they affect model performance. This lack of rationale may undermine the scientific validity and reproducibility of the method. The authors should provide theoretical or experimental support for the choice of hyperparameters to enhance the paper's credibility and rigor.
5）Lack of Explanation for Experimental Results:
The experimental results show that the proposed method is not optimal in certain settings, but the paper does not provide an in-depth analysis of these suboptimal outcomes, such as the reasons for performance degradation on specific datasets or model combinations. This lack of discussion may weaken the paper's comprehensive understanding of the method's strengths and weaknesses. The authors are encouraged to supplement detailed analyses of these results and explore directions for improvement to enhance generalizability and stability.

**Questions:**

3. Weaknesses and Questions
1）Unclear definition of terms and objectives:
The definition of "Dynamic Graph Unlearning" and the specific goals of unlearning requests are vague, especially regarding the distinction from static graph unlearning. This might hinder readers' understanding of the unique challenges and contributions of dynamic graph unlearning.
Key concepts, such as the theoretical foundation of "gradient transformation" and its applicability in various scenarios, lack sufficient elaboration, leading to potential confusion.
2）Clarification of Methodological Limitations:
The paper attributes the limitations of prior methods to their reliance on preprocessing, but this is not entirely accurate. The main issue is that these methods cannot support the dynamic update requirements of DGNNs and are only applicable to static scenarios during the early stages of development, lacking support for subsequent dynamic unlearning tasks. The authors are advised to revisit and accurately describe the true limitations of previous methods to enhance the rigor of the paper.
3）Insufficient discussion on practical deployment feasibility:
While the method demonstrates advantages in resource efficiency and time performance, potential challenges in practical deployment, such as adaptability in distributed environments and computational requirements on edge devices, are not sufficiently discussed. The impact of computational bottlenecks and latency in high-frequency unlearning request scenarios requires further analysis.
4）Lack of Explanation for Hyperparameter Settings:
The hyperparameter settings in Equation (11) lack detailed justification, and it is unclear why the current values were chosen and how they affect model performance. This lack of rationale may undermine the scientific validity and reproducibility of the method. The authors should provide theoretical or experimental support for the choice of hyperparameters to enhance the paper's credibility and rigor.
5）Lack of Explanation for Experimental Results:
The experimental results show that the proposed method is not optimal in certain settings, but the paper does not provide an in-depth analysis of these suboptimal outcomes, such as the reasons for performance degradation on specific datasets or model combinations. This lack of discussion may weaken the paper's comprehensive understanding of the method's strengths and weaknesses. The authors are encouraged to supplement detailed analyses of these results and explore directions for improvement to enhance generalizability and stability.

**Reviewer Confidence:**

2: The reviewer is willing to defend the evaluation, but it is likely that the reviewer did not understand parts of the paper

**Scope:**

4: The work is relevant to the Web and to the track, and is of broad interest to the community

---

### Official Review · Reviewer_jvaN · 2024-11-30

**Novelty:** 3
**Technical Quality:** 4

**Review:**

Summary of the Paper:
This submission introduces a graph unlearning method called Gradient Transformation in the setting of dynamic graphs, which can be used in a post-processing manner. The approach is evaluated on six datasets and two DGNN architectures, demonstrating strong performance gains in efficiency and robustness.

Strength:
The submission is well-written and easy to follow.
The development of ideas and presentation (with figures) is mostly clear and easy to follow.
The experiments are extensive and explore interesting questions.

Weakness:
Dynamic graphs can be regarded as static graphs arranged according to time step, so the traditional static machine unlearning technology is still valuable to discuss in the field of dynamic graphs. It is hoped that the authors can do more relevant experiments and make further discussions.
In the field of machine unlearning, the model usually does not need to use the validation dataset, but the purposed method requires the validation dataset. If the existence of the validation set is removed, whether the method is still valid is worth discussing.
The training set and validation set are i.i.d in this paper, but performance may be compromised if they are Non-i.i.d. The authors can add more discussions.
In Table 6, it can be found that the performance improvement of Ours-re-ul-reg and Ours-full is not very significant, and the accuracy of the remaining dataset and the AUC in the test dataset are even declined. Hope that the author can analyze its reasons in detail.

**Questions:**

Please review the weakness outlined above.
The authors refer to the hyperparameter setting of Eq. 11 in Section 6.1. Could the authors provide the experimental results with different parameter settings to support the reasonableness and validity?
Could the authors explain why the two-layer MLP-Mixer was served as the unlearning model instead of other more complex architectures?

**Reviewer Confidence:**

3: The reviewer is confident but not certain that the evaluation is correct

**Scope:**

4: The work is relevant to the Web and to the track, and is of broad interest to the community

---

### Official Review · Reviewer_PZ82 · 2024-12-01

**Novelty:** 5
**Technical Quality:** 5

**Review:**

This manuscript is the first to study the unlearning problem in dynamic graph neural networks (DGNN) and proposes a gradient based post-processing method to obtain desired parameter updates.

**Questions:**

1. In Preliminaries,  the manuscript mentions that the message function for edge events is designed as \( m_i \) and \( m_j \). What is the relationship and difference between \( m_i \) and \( m_j \)?
2. The manuscript mentions that S is a dynamic graph, and Sul is a subset of S. However, in the example on lines 302-303 of the manuscript, Sul appears to be a subset of the update events？Normally, a dynamic graph should be different from an event sequence.
3. On line 388 of the manuscript, the variable Y is introduced in the formula. Why is Y used here instead of the image and the previously mentioned At? And what does Y represent?
4. On line 402-403 of the manuscript, both the input and the output are represented as \(\bigtriangleup \theta\). Is this appropriate?

Some minor problems:
1. In formula (2), why is "I" used between the adjacency matrix \( A_t \) and \( f(s) \)?
2. In the paper, many formulas use a font other than Times New Roman.
3. In the manuscript, GeLU is referred to as an "active function", should it be "activation function" instead?

**Reviewer Confidence:**

3: The reviewer is confident but not certain that the evaluation is correct

**Scope:**

4: The work is relevant to the Web and to the track, and is of broad interest to the community